# Predicting Embryo Morphokinetics in Videos with Late Fusion Nets & Dynamic Decoders

**Nathan H Ng & Julian McAuley**
University of California, San Diego
{nhng, jmcauley}@ucsd.edu

**Julian A Gingold & Nina Desai**
Cleveland Clinic Foundation
{gingolj, desain}@ccf.org

**Zachary C Lipton**
Carnegie Mellon University
zlipton@cmu.edu

## Abstract

To optimize clinical outcomes, many fertility clinics select embryos strategically, based on how quickly they reach certain developmental milestones. This requires manually annotating time-lapse EmbryoScope videos with their corresponding morphokinetics, a time-consuming process that requires experienced embryologists. We propose late-fusion ConvNets with a dynamic programming-based decoder for automatically labeling these videos. Experiments address data extracted from EmbryoScope incubators at the Cleveland Clinic Foundation Fertility Center. We focus on 6 stages, demonstrating 87% per-frame accuracy.

## 1 Introduction

To select embryos for in vitro fertilization (IVF) (Cetinkaya and Kahraman, 2016), doctors typically rank them based on a morphological evaluation. Due to advancements in incubators with built-in time lapse monitoring (EmbryoScope), morphokinetics can be determined non-invasively (Chamayou et al., 2013). These incubators use a high powered microscope to capture images of a developing embryo every 15 minutes. Embryologists perform the analysis manually, viewing a sequence of photographs and annotating the time stamps at which each embryo achieves certain milestones. These scores are combined according to a heuristic formula to rank the embryos by *believed* viability for transfer into a prepared endometrium.

We investigate deep learning techniques for automating the labeling of these transition times. Specifically, we consider different methods of incorporating the temporal context of the video into the model architecture by fusing frame information earlier vs later on. We also consider the use of a post-processing decoder to enforce monotonicity constraints on our predictions.

Our dataset consists of 1309 EmbryoScope time lapse videos extracted from incubators at the Cleveland Clinic Foundation Fertility Center. Each frame is a grayscale $500 \times 500$ resolution image with a *well number* in the lower left corner and an hour time marker in the lower right corner, as seen in Figure 1a. The videos in our dataset span 113 different patients, each with up to 12 wells and corresponding videos. Videos begin around 18 hours after fertilization, and end around 140 hours after fertilization. For each video, we obtained manually labeled times at which the depicted embryo was first observed at each developmental stage. In reality, these stages are monotonically increasing; once an embryo reaches a later stage, it cannot regress to an earlier one.

We focus on 6 stages of development for each embryo, cutting off each video at hour 70. The stages include the initial stage (tStart), the appearance and breakdown of the male and female pronucleus (tPNf), and the appearance of 2 through 4+ cells (t2, t3, t4, t4+). The class distribution of frames are 10.3%, 5.3%, 19.4%, 4.5%, 19.8%, and 40.7% in each stage, respectively (Figure 1b gives a fuller summary of stage distribution). We create training/validation/test splits by randomly selecting 93/10/10 patients and their respective wells. This gives us a total of 191449 / 21798 / 23348 frames in the respective splits.

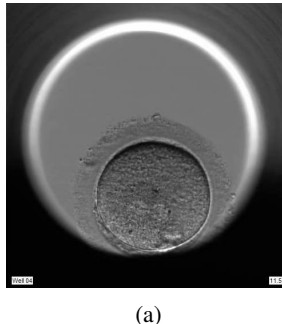
(a)

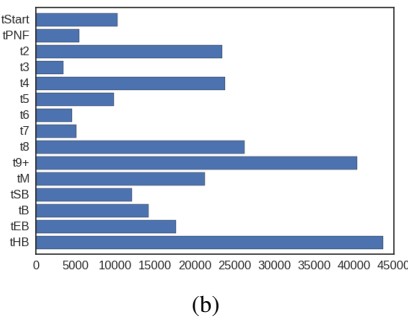
(b)

Figure 1: Left (a) sample frame from an EmbryoScope (public image), and right (b) summary statistics on the number of frames assigned to each stage of development in human annotations.

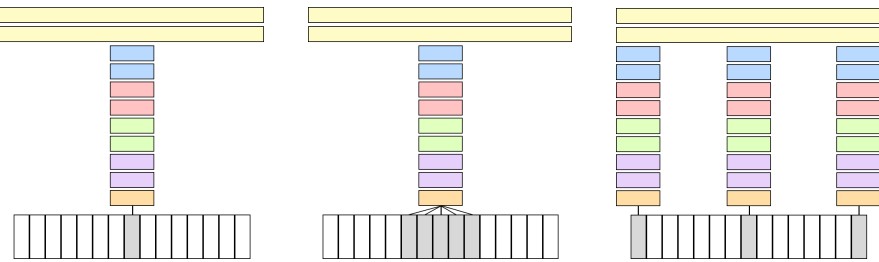

Figure 2: Model architectures: single (left), early-fusion (center), late-fusion (right)

## 2 METHODS

We cast predicting embryo morphokinetics as a multiclass classification problem, where the input is an EmbryoScope video, and the output is a label indicating which stage the embryo is currently in at each frame of the video. Because the frames arrive in sequence, and we have prior knowledge that human embryologists rely on contextual frames (several before, several after) to make their annotations, we investigate several methods of incorporating the context frames and temporal information. Among other techniques, we (i) consider the use of time as a feature and (ii) experiment with various architectures for incorporating context from temporally local frames.

We first consider a baseline of a single frame ResNet (He et al., 2016) architecture to understand how predictive static images are of morphokinetics. We then consider two methods of temporal fusion: *early fusion* and *late fusion* (Karpathy et al., 2014) (Figure 2). In early fusion, data is combined temporally at a pixel level. This is accomplished by stacking surrounding frames prior to applying any convolutions and modifying the number of channels in the input of our single frame model. In late fusion, all frames in the context window are encoded via identical ConvNets. We concatenate the resulting representations and pass them through the fully-connected layers to generate outputs.

**Structured Decoding with Dynamic Programming (DP)** The ground truth stages must be monotonically non-decreasing. A vanilla CNN does not have this constraint. We impose this inductive bias through a dynamic programming postprocessing that enforces monotonicity of predictions. For any video, our model outputs probabilistic predictions $\hat{\boldsymbol{p}}^t$ at every frame $t \in 1, ..., T$, where $T$ is the last frame and $\hat{p}_i^t$ is the estimated probability that frame $t$ is in stage $s_i$. We define $E(\hat{\boldsymbol{y}}, \hat{\mathbf{P}})$ as an expected cost for final predictions $\hat{\boldsymbol{y}}$ given the model outputs $\hat{\mathbf{P}}$ (over all frames $t$). Both losses are sums of expected per-frame losses $\sum_{t=1}^{T} e(\hat{y}^t, \hat{\boldsymbol{p}}^t)$, which we must optimize subject to a monotonicity constraint: $\hat{y}^{t+1} \geq \hat{y}^t, \forall t$. We investigate two per-frame losses, negative label likelihood (LL) and earthmover (EM) distance, defined by $e_{LL}(\hat{y}, \hat{\boldsymbol{p}}) = -\log(\hat{p}_{\hat{y}})$ and $e_{EM}(\hat{y}, \hat{\boldsymbol{p}}) = \sum_i \hat{p}_i |\hat{y} - i|$, respectively. Because there are only $T$ frames, and 6 possible choices of the label, our DP has $6T$ subproblems. DP is required because once we choose to advance the prediction $\hat{y}^t$ to stage $i$ we impact the loss at all subsequent steps owing to monotonicity.

Table 1: Quantitative results for various architectures and output decoding schemes

| Model | Frames | Raw Acc. | DP: label likelihood s.t. monotonicity | | | DP: earthmover's distance s.t. monotonicity | | |
|---|---|---|---|---|---|---|---|---|
| | | | Accuracy | MAE | RMSE | Accuracy | MAE | RMSE |
| ResNet50 | 1 | 0.8200 | 0.8460 | 11.225 | 29.650 | 0.8368 | 11.115 | 28.899 |
| Early Fusion | 3 | 0.8237 | 0.8448 | 10.555 | 27.370 | 0.8397 | 10.687 | 27.962 |
| Early Fusion | 9 | 0.8252 | 0.8423 | 10.927 | 29.400 | 0.8362 | 10.808 | 28.375 |
| Early Fusion | 15 | 0.8182 | 0.8456 | 10.935 | 27.719 | 0.8364 | 11.242 | 27.756 |
| Early Fusion+time | 9 | 0.8343 | 0.8430 | 11.152 | 26.321 | 0.8388 | 11.068 | 25.979 |
| Early Fusion+time | 15 | 0.8420 | 0.8446 | 10.761 | 26.849 | 0.8411 | 10.904 | 26.701 |
| Late Fusion | 15 | **0.8479** | 0.8676 | 8.963 | 24.756 | **0.8708** | **8.594** | **24.334** |

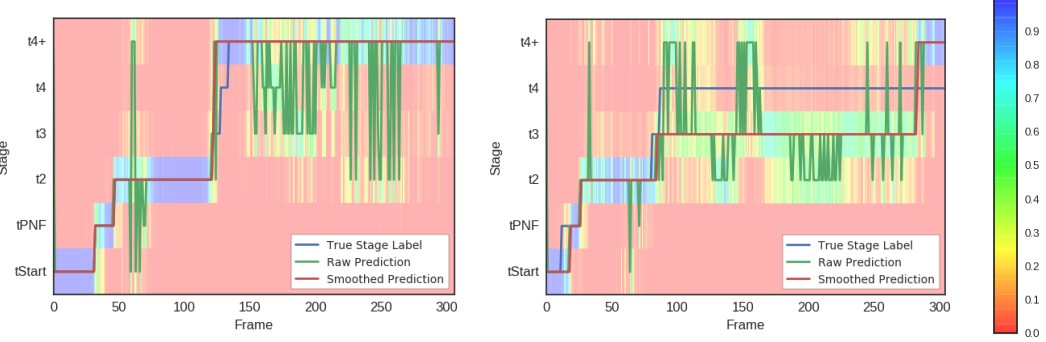

Figure 3: DP decoders smooth predictions. On left, smoothing significantly reduces the error caused by the model's uncertainty in later stages.

## 3 EXPERIMENTS

We now present our basic experimental setup. We report the per-frame accuracy of our raw predictions, as well as the per-frame accuracy of the DP predictions (for both objectives). We also report the mean absolute error (MAE) and root mean squared error (RMSE) (measured in frames) of the predicted transition times after post-processing.

Our models are variants of a ResNet50 pretrained on ImageNet(He et al., 2016). For vanilla and early-fusion models, we modify the number of input channels to take in varying amounts of temporal context. Our late fusion model concatenates the fully-connected layers together after applying parallel convolutional encoders separately. In some experiments, we append the timestamp as an input to the fully-connected layer. We train all models using stochastic gradient descent with weight decay and use standard data augmentation techniques, including random rotations and flips and choose all hyper-parameters based on validation set performance. Results (Table 1) demonstrate that our best model, by all metrics, is a late-fusion ResNet with outputs decoded to minimize the expected earth mover's distance.

## 4 RELATED WORK

The problem of predicting embryo annotations from time lapse videos has been addressed in the literature by Khan et al.. In that work they use an 8-layer convolutional network to count the number of embryos in an image. To incorporate temporal information they use conditional random fields and similarly use dynamic programming to enforce monotonicity constraints. To our knowledge, our work is the first to use deep learning to predict embryo morphokinetics and to model it using late fusion models and dynamic decoders. The idea of extending models to include temporal information has been explored extensively in recent years. Simonyan and Zisserman (2014) used a two-stream architecture applied to a single frame as well as multi-frame optical flow in order to combine spatial and temporal information. Karpathy et al. (2014) introduced the architecture of early, late, and slow fusion models in order to incorporate temporal information at different points in the model.

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
