# OpenReview forum: "Predicting Embryo Morphokinetics in Videos with Late Fusion Nets & Dynamic Decoders"
_ICLR.cc/2018/Workshop — Accept_

### Official Review · AnonReviewer2 · 2018-03-10
**CNN + DP based postprocessing for embryo morphokinetics prediction**

**Rating:** 6
**Confidence:** 3

**Review:**

The paper proposes deep learning based methods for predicting embryo morphokinetics, as a multiclass classification problem with temporal constraints of monotonicity.

With a baseline of single frame based ResNet50, they add temporal constraints i.e. monotonicty of label occurrence with time using DP based postprocessing.

With the recent ResNet architecture they achieve ~84% frame level accuracy on a significant sized dataset with 200k train and 20k val and test frames each resp. With the addition of temporal constraints this increases to 87% (with 6 stages of embryo development).

As the authors mentioned, the problem has been addressed before; this work is the first one to report deep learning based results which are reasonably high at 87% (although I do not know if these results are practically useful yet).

---

### Official Review · AnonReviewer1 · 2018-03-18

**Rating:** 6
**Confidence:** 4

**Review:**

This paper focuses on the problem of predicting the timestamps at which embryos reach certain developmental checkpoints (morphokinetics) in in-vitro fertilization imagery. The basic idea is that a base image classifier, ResNet, is trained to recognize the moprhokinetics from single images.  However, predictiong in single images means that subsequent frames might be labelled differently, and also that later on frames might receive labels that refer to earlier developmental stages than previously predicted, implying developmental regression, which is not "allowed" in this dataset. For this reason an intelligent post-processing step is proposed based on dynamic programming to correct for prediction inconsistensies.

The problem is interesting and practical. The solution proposed is a logical one, although not great technical contribution can be claimed. That is still ok, posing the question is more relevant than answering one, for such a problem.

The main problem of this paper is that it does not discuss results. Instead, the results are merely aggregated in a table, however, there is no analysis. I would expect the authors to be more detailed in this aspect.

- Interesting problem
- If the problem is not modelled as sequence classification, can't the model predict different labels within a sub-sequence, thus indicating "embryo regression"?
-

---

### Decision · Program_Chairs · 2018-03-20
**ICLR 2018 Workshop Acceptance Decision**

**Decision:**

Accept

**Comment:**

Congratulations, your paper was accepted to the ICLR workshop.